# Capsule Endoscopy for Gastric Evaluation

**DOI:** 10.3390/diagnostics11101792

**Published:** 2021-09-28

**Authors:** Ji-Hyun Kim, Seung-Joo Nam

**Affiliations:** School of Medicine, Kangwon National University, Chuncheon 24341, Korea; kimjihyun81@naver.com

**Keywords:** wireless capsule endoscopy, stomach, magnetic control

## Abstract

Wireless capsule endoscopy was first developed to observe the small intestine. A small capsule can be swallowed and images of gastrointestinal tract are taken with natural movement of peristalsis. Application of capsule endoscopy for observing the stomach has also received much attention as a useful alternative to esophagogastroduodenoscopy, but anatomical characteristics of the stomach have demanded technical obstacles that need to be tackled: clear visualization and active movements that could be controlled. Different methods of controlling the capsule within stomach have been studied and magnetic manipulation is the only system that is currently used in clinical settings. Magnets within the capsule can be controlled with a hand-held magnet paddle, robotic arm, and electromagnetic coil system. Studies on healthy volunteers and patients with upper gastrointestinal symptoms have shown that it is a safe and effective alternative method of observing the stomach. This work reviews different magnetic locomotion systems that have been used for observation of the stomach as an emerging new application of wireless capsule endoscopy.

## 1. Introduction

Wireless capsule endoscopy (WCE) was first introduced in 2000 as a novel and least invasive modality for visualization of the gastrointestinal (GI) tract [1], specifically the small bowel mucosa [2]. Flexible fiberscopes were the only available method of observing the stomach or colon and the effort to observe the small bowel, which could not be reached by conventional endoscope, led to the development of WCE. WCE is composed of four components, including the wireless capsule endoscope that is swallowed, the data-receiving box that receives images transmitted from the capsule endoscope, the working station for analysis of images obtained, and the application software [1,3]. Since then, WCE for endoscopic evaluation of the esophagus and colon has also been developed, which is commercially available today. WCE has received much attention as a useful alternative to esophagogastroduodenoscopy (EGD, the standard diagnostic modality for gastric lesions) because EGD is uncomfortable for patients. In addition, EGD performed under sedation may cause adverse events in high-risk patients [4]. Compared to EGD, WCE involves swallowing a small capsule and the camera within the capsule captures images of the entire GI mucosa with natural peristaltic movements. Thus WCE can be an effective and safe alternative to conventional EGD, particularly during the ongoing coronavirus disease (COVID-19) pandemic as it can reduce the risk of exposure to aerosols generated during EGD [5]. However, the development of WCE for the stomach has been a challenge because, unlike the esophagus, small intestine, and colon, which are anatomically long cylindrical structures, the stomach is a capacious organ with a unique anatomy, and mucosal visualization is difficult in a collapsed stomach in the fasting state. Moreover, movement of WCE relies on natural peristaltic movement of the GI tract, which may not be a problem when observing the small bowel or colon, but this can interfere with complete and accurate observation of the entire gastric mucosa [6].

In order to develop effective WCE for observation of the stomach, technical obstacles included development of an active locomotion system that enables accurate movement to specific targets against peristalsis, orientation within the stomach, and method of expanding the stomach. Locomotion systems that control capsule movements had been integrated into the capsule to ensure complete examination [7,8]. Capsules with internal mechanical locomotion systems include legged locomotion [9,10,11], worm-like locomotion [12,13], and paddling-based systems [14]. However, it is difficult to translate such technology into real-world clinical practice because internal locomotion requires a large amount of power that cannot be compacted into small capsule, and mechanical locomotion itself occupies a large component of the volume [6,15]. Magnetic actuation has emerged as an innovative technology to control capsule movements. One of the early applications of the magnetic field as a guidance system for movement of a medical device was in the manipulation of a steel-tip catheter inside a blood vessel [16]. Experimental study on the application of a magnetic system for manipulation of a catheter within the brain has also been reported [17] and a magnetic navigation system for controlling catheter movements during ablation procedures for arrhythmia are already in commercial use [18]. This type of locomotion allows miniaturization of capsules by reducing their power demands and dependence on the internal mechanical system. Magnetically controlled CE (MCE) is the only locomotion system that has been applied to human research with results from clinical data. Several studies have shown that the diagnostic accuracy of MCE is similar to that of standard EGD [19,20], which supports the role of MCE as a novel, minimally invasive method to screen the human stomach. This review focuses on studies that describe the clinical applications of MCE as an advanced method for endoscopic visualization of the stomach.

## 2. Magnetic Actuation

The magnetic actuation system consists of a capsule shell with magnetic material, which can be created by mixing magnetic powder of neodymium-boron-iron with silicone [21], or a capsule modified by placing magnetic material at one end. The movement of this modified capsule can be controlled by an external magnet located outside the body [6]. The magnetic field may be generated by this external device, such as a permanent-type hand-held magnet and robotic arm or by electromagnets that generate a variable degree of magnetic field. The magnetic field produces rotational and translational forces that enable movement, velocity control, orientation, localization, and accurate imaging [8]. This system uses a device that operates outside the body and represents an external locomotion system, which was the original model tested in humans [22]; therefore, all clinical studies that report the use of this technology refer to this type of magnetic actuation [7]. Different types of magnetic actuation are presented in Figure 1. Commercially available systems are summarized in Table 1 and studies on different magnetic actuation have been summarized on Table 2, Table 3, Table 4 and Table 5.

## 3. Magnetically Guided Capsule Gastroscopy Using Hand-Held Magnets

Swain et al. reported the first human study of wireless CE using a hand-held magnet for visualization of the stomach [22]. Using a colon-type capsule (Given Imaging Ltd., Yoqneam, Israel) modified by placing magnetic material at one end of the dome and programmed to transmit images at the rate of 4 frames/s, the authors tested the performance of the capsule on one volunteer. A paddle-shaped hand-held magnet was placed on the chest and abdomen to manipulate capsule movement. An esophagogastroduodenoscope was inserted to observe capsule movements, and the images captured were viewed in real time together with the EGD images. The magnet was successfully held and manipulated through the esophagus for 10 min with complete evaluation of the Z-line and was successfully maneuvered thereafter along the stomach axis. The capsule was rotated, held, and moved with the patient in the supine position without any discomfort caused by capsule movement. This was the first study to suggest the potential application of MCE for visualization of the stomach in humans; however, this approach was associated with the following limitations: (i) upward and downward movements of the capsule within the esophagus were difficult owing to significant distance between the external magnet and the magnet within the capsule and, (ii) it was difficult to identify the direction of the capsule without EGD-guided capsule visualization. Air insufflation performed during EGD could have ensured better capsule maneuverability.

Similarly to the study reported by Swain et al., Keller et al. [23] used a hand-held magnet to maneuver the capsule in 10 healthy volunteers; the authors used a colon capsule (Given Imaging Ltd.) with a magnetic disc inserted into one dome, which was programmed to transmit 4 frames/s. Abdominal magnetic resonance imaging (MRI) was performed prior to MCE to confirm the distance between the magnets. Following capsule ingestion and its entry into the stomach, participants were administered sherbet powder, which released carbon dioxide to distend the stomach. Tolerability was determined using questionnaires for evaluation of pain, swallowing difficulties, and other complaints. Safety was evaluated based on any adverse events and performance through visualization and manipulation. Only one participant developed a pressure sensation during the study. Capsule movement towards even small targets was identified in seven participants, and 75–90% of the gastric mucosa was clearly observed in seven patients. The study showed excellent maneuverability and visualization of the gastric mucosa in most subjects, and this technique was deemed safe and well tolerated. However, capsule movement from the proximal to distal stomach necessitated a change in the subject’s position, and large amounts of opaque fluid could not be cleared.

In 2016, Rahman et al. [24] used the MiroCam-Navi (Intromedic Ltd., Seoul, South Korea) navigation system (Figure 1a) with an accompanying hand-held magnet in 26 volunteers to test the degree of visualization and maneuverability. From a prior study, the authors had identified four important locations for placing the hand-held magnet that would provide optimal visualization [44]. Following the ingestion of metoclopramide, Pronase, and simethicone to aid visualization and gastric contraction, participants were administered the capsule and a nasogastric tube was inserted after the capsule had entered the stomach to identify the cardia, which may be difficult to locate using the capsule alone. Participants did not report significant discomfort upon completion of the procedure. The study results showed that visualization and imaging (obtaining clear views) differed across different parts of the stomach. Optimal visualization, defined as a non-obscure view that enabled clear identification of landmarks and mucosa, was best in the stomach body (100%) and poorest at the cardia (88%); notably, clear views were best obtained following optimal pre-procedural stomach preparation. The capsule identified erosions and gastritis in four patients each, and additional standard EGD confirmed these lesions. However, EGD successfully diagnosed a 5 mm sized submucosal lesion, which was missed by the capsule owing to suboptimal preparation and difficulty with obtaining clear views of the cardia. Moreover, maneuverability was poor in the proximal stomach because the distance between the magnet and the ventral surface was greater than that between the magnet and distal stomach (16.5 cm vs. 9 cm). The proximal stomach is more capacious; therefore, light from the capsule is dissipated over a larger area, and additionally, opaque gastric contents prevent optimal visualization of the proximal stomach.

In contrast to previous studies that included healthy volunteers, Ching et al. [25] used the MiroCam-Navi system in 49 patients with iron deficiency anemia to identify pathological lesions in the stomach and compared their findings with those obtained after EGD. Following the ingestion of 1 L of water to distend the stomach, participants were administered the capsule, which was manipulated using a hand-held magnet. The study reported complete visualization of the cardia in 95.9% and of the body in 98% of patients. Compared with other studies, patients underwent CE for small bowel examination and were, therefore, administered polyethylene glycol 3350 (KLEAN-PREP^®^) the evening before the procedure. Visualization of all parts of the stomach was excellent, defined as a 100% complete view of the landmarks, except for the fundus, which showed <50% visualization owing to debris, bubbles, or poor clarity. The detection rate of gastric lesions was higher with MCE than with EGD (36 vs. 5, *p* < 0.001). MCE was able to accurately detect gastritis, gastric ulcers, gastric angiodysplasia, gastric polyps, and hiatal hernia and was well tolerated by patients.

In another study, Ching et al. [26] used the MiroCam-Navi system to evaluate the performance of MCE. This was a prospective cohort study that included 33 patients with suspected acute upper GI bleeding. Results from MCE were compared to EGD findings. No statistically significant difference was observed in the detection of significant lesions that are likely causes of bleeding by MCE (peptic ulcers, esophageal varices, and gastric varices) and those detected by EGD (14 vs. 13, *p* = 1), which identified esophageal varices, gastric varices, gastric ulcers, and duodenal ulcers. MCE could identify lesions missed by EGD and these included esophageal and duodenal bulb ulcers in one and four patients, respectively. Analysis of lesions, including the non-significant ones, showed that MCE could detect more lesions than EGD (82 vs. 49, *p* = 0.0004) and the visual analogue scale scores for pain and discomfort were significantly lower than those in patients who underwent unsedated EGD. A complete view (defined as clear identification of landmarks) was obtained in the esophagus, cardia, greater and lesser curvatures, anterior and posterior body walls, antrum, and pylorus, whereas a reasonable view (defined as <50% visualization owing to debris, bubbles, or poor imaging) was observed in the fundus and first part of the duodenum. These results suggest the usefulness of MCE as a potential tool to delineate the etiology of upper GI bleeding.

Lien et al. [27] described modified MCE and a hand-held external magnetic field navigator. In contrast to a wireless capsule device used in previous studies, the authors introduced CE using a wire attached to one end of the capsule for transmission of images (30 frames/s) and power. In vitro and ex vivo studies showed that, together with an external magnetic field navigator, this approach was associated with high-precision rotation and controllable movements of the capsule [45]. The same device was tested in nine healthy volunteers [27], who ingested approximately 500 mL of water before the test and subsequently swallowed the capsule, and a hand-held magnet was placed on their abdomen to control capsule movement. The participants were administered an air-producing powder to distend the stomach. Excellent mucosal visualization (defined as no mucus adherent to the mucosa) was achieved in three patients, good visualization (defined as a small amount of mucus that adhered to the mucosa but did not obscure the view) was achieved in three patients, and fair visualization (defined as a small amount of mucus adherent to the mucosa with a partially obstructed view) was achieved in three patients. Complete obstruction of the view was not encountered in any patient. Approximately 75–100% visualization was achieved in the fundus to the pylorus, and distension of the stomach was better observed in the distal stomach. Capsule movement from the cardia to the pylorus, reverse movement at the antrum and duodenum, and backward movement from the antrum to the cardia were all observed in five patients (55.6%). With regard to tolerability, epigastric discomfort and nausea occurred in one patient each; however, no severe adverse events were reported. The authors observed that it was possible to capture a greater number of images at specific sites because a greater number of frames were captured per second, and this method facilitated accurate manipulation owing to the traction of the cable. Lin et al. [28] performed MCE using this same device in 15 elderly patients with gastric symptoms (dyspepsia, acid reflux, heartburn, epigastric pain, or anorexia). Owing to its portability, out-of-hospital MCE could be performed and home visits were possible. Complete observation of gastric landmarks was achieved in 81.25% of patients; however, the cardia and fundus were not completely visualized in 18.75% of patients. The authors attributed these limitations to bubbles and inadequate gastric preparation, patients’ difficulty with changing positions, or ingestion of additional water. The procedure was well tolerated by patients, and all patients completed the examination without discomfort or other complications necessitating termination of the procedure. This study suggested the feasibility of bedside MCE in patients who were unable to visit the hospital.

MCE performed using an external hand-held permanent magnet is non-invasive and safe, and portability of the device enables out-of-hospital WCE. Studies on hand-held magnets are summarized in Table 2. Patients are instructed to undergo standard EGD if MCE detects abnormalities that necessitate additional testing. This system is similar to the standard EGD procedure because the operator receives visual input and adjusts capsule location or movements in response. However, the operator does not receive feedback regarding the magnetic strength, which cannot be adjusted and is required to manually move the hand-held magnet to achieve the desired motion [6]. This procedure requires technical skill, and a definite learning curve is observed in clinical practice; therefore, the operator must receive appropriate training to improve capsule maneuverability, which is similar to the training and learning curve associated with EGD. Another drawback is the weight of hand-held magnet paddle. Magnets with larger force are heavier to hold, and if examination time is extended it can cause fatigue. Furthermore, observation of the proximal stomach is more challenging because of the greater distance between the fundus and the ventral surface compared with that between the antrum and ventral surface (16.5 cm vs. 9 cm) [24]. Larger magnets are required in patients with obesity, which is technically demanding for the operator [8].

## 4. Magnetically Guided Capsule Gastroscopy Controlled by an Electromagnetic Coil System

The MCE developed by the Olympus Medial Systems Corporation (Tokyo, Japan) and Siemens Healthcare (Erlangen, Germany) uses an electromagnetic coil system similar to existing MRIs with lower magnetic power to provide varying magnetic field vectors and advanced motion schemes (Figure 1b) [6].

Rey et al. [29] used this system to perform MCE. The system uses a maximum magnetic field of 100 mT, which is 15-fold smaller than that of the standard 1.5 T MRI. The capsule contains two image sensors that transmit images at 4 frames/s, which are displayed live on monitors. The operator can navigate the capsule using two joysticks for movement at 5 degrees of freedom, including tilting and rotating along different axes. The study included 29 volunteers and 24 patients with symptoms of epigastric pain or reflux; no pre-medications were administered and the participants ingested 1300 mL of water. Complete mucosal visualization was achieved in 73–98% of patients depending on the location of the stomach; the poorest and best visualization were achieved in the fundus and antrum, respectively. Compared with the findings of conventional EGD, 30 lesions were diagnosed in this study; 14 lesions were detected by both MCE and EGD, 6 additional lesions only by EGD, and 10 only by MCE. Lesions identified by MCE but missed on EGD included polyps, inflammation, angiodysplasia, and ulcers, which suggest that the newly developed MCE technique using a controlled magnetic field may serve as an innovative, non-invasive method for visualization of the stomach. Rey et al. [30] used the same system to evaluate the performance and maneuverability of the device in 61 patients with indications to undergo EGD. The patients ingested the capsule in the sitting position and were subsequently placed in the supine position. Similarly to a previous study that used the same protocol and device, the capsule could be maneuvered to move to the water surface or sink to the bottom of the stomach. Images were magnified secondary to refraction in water, and the reported complete visualization rate was 85.2–93.4% across various parts of the stomach. The mean evaluation time decreased from 30 min in the previous study to 17.4 min in this study, which suggests that the technique utilized in this study required less time as the operator gained greater experience. Similarly to a previous study, 108 lesions were identified, of which 63 were detected by both EGD and MCE, 14 additional lesions missed by MCE were detected by EGD, and 31 lesions missed by EGD were detected by MCE. Upon completion of MCE, only one patient reported abdominal pain that spontaneously resolved, which indicates that the procedure was well tolerated.

In contrast to previous studies that included healthy volunteers, Denzer et al. [31] investigated 189 patients with upper GI symptoms and compared the results of MCE with EGD. The magnetic guidance system was similar to the model used in a previous study [29] with the same capsule and same pre-procedural preparation. Lesions were classified as major or minor; major lesions were defined as those with therapeutic relevance that necessitated biopsy or therapy (adenomas, carcinomas, polyps, ulcers, and angiodysplasias), whereas minor lesions were defined as multiple diffuse findings including fundic gland polyps, erosions, and atrophy. Only 21 (11%) patients had major lesions, two of which were adenocarcinomas. The accuracy of MCE for detection of major and minor lesions was 90.5% and 88.1%, respectively. No complications were observed, and all patients indicated that they were willing to undergo subsequent MCE evaluation. This study is interesting because it included a relatively large number of patients who underwent MCE evaluation that accurately detected major lesions such as adenocarcinomas, which represent one of the most common indications for screening EGD. The study also highlights the accuracy of MCE for detection of lesions that require further testing by EGD and suggests the potential usefulness of this system as a routine screening tool in symptomatic patients. Although MCE in detecting major lesions had high specificity of 94.1%, the sensitivity of MCE was only 61.9%, which suggests major improvement is needed before using it as a screening test to identify patients requiring further EGD. One of the major limitations of this device was limited expansion of stomach. Unlike conventional EGD, which uses air for stomach expansion and maintenance of the expanded state, the water ingested during MCE left the stomach too quickly, hindering accurate examination of single focal lesions.

As opposed to a hand-held magnet, this type of MCE can generate dynamic magnetic fields, and the capsule can be moved along 5 degrees of freedom with tilting and rotation equivalent to the large and small wheel movement, respectively, used during EGD. Studies on electromagnetic coil systems have been summarized in Table 3. Although MCE using MRI system offers dynamic magnetic fields in contrast to hand-held magnets, it requires installation, which is expensive.

## 5. Magnetically Guided Capsule Gastroscopy Controlled by a Robotic Arm

An animal study that compared manual and robotic control for magnetic steering of CE showed that robotic control is more accurate and reliable [46]. Similarly to MRI systems, the system consists of a controller that has an examination bed, a magnetic head generating a magnetic field, and a translational rotational platform for linear and rotational movement of the external magnet. After ingesting the capsule, the patient lies down on the examination bed and the magnetic head is controlled by joystick movements to facilitate movement of capsule within the stomach. It is cheaper compared to MRI systems, and the magnet movement is controlled by a robotic arm, which provides accurate movement without fatigue on the part of the operator. Also, automatic movement is possible.

Liao et al. [32] investigated the role of a robotic-controlled magnetic CE (Ankon Technologies Co, LTD) in 34 healthy volunteers. The robotic guidance system consists of a C-arm-type device (Figure 1c), which offers high levels of accuracy and precision. The system also provides 5 degrees of freedom (translational and rotational), and the capsule transmits images at the rate of 2 frames/s for each sensor. The participants ingested approximately 1 L of water and 6 g of air-producing powder before they ingested the capsule. Following capsule ingestion, they were placed in the supine position on a bed attached to the guidance robot. Previous studies have described that participants were frequently required to change positions; however, in this study, they remained supine with minimal movement. The operator controlled the capsule motion by lifting, rotating, and maneuvering it toward different regions of the stomach, and, following examination, all participants completed a questionnaire to evaluate any discomfort experienced during the study. Gastric preparation was good in most participants (88.2%) with transparent fluid, and > 95% of the mucosa was clearly visualized. Adequate gastric distension was achieved in 85.3% of the participants, and only a few gastric folds remained undistended. Good maneuverability, defined as smooth movements with accurate detection of lesions, was observed in 85.3% of participants, and accurate capsule movement was difficult in 14.7% of participants, which was attributable to a high body mass index and moderate visualization. Tolerability was good among all participants, and only one participant reported slight abdominal distension after ingestion of the air-producing powder. This pilot study suggested the usefulness of MCE controlled by a robotic arm, a novel modality for stomach evaluation.

Using the same system, Zou et al. [20] performed a comparative trial at two tertiary centers. Using an identical method of observation based on a study by Liao et al. [32], the authors compared the findings between 68 patients who underwent MCE and EGD. Based on result interpretation, evaluation was categorized as normal or abnormal (mild inflammation was also classified as normal); 48 patients were categorized as showing abnormal findings on both MCE and EGD, which indicates an overall agreement of 91.2% (95% confidence interval [CI] 84.4–97.9%, *p* = 0.687) with no statistically significant differences between the findings detected by the two modalities. A total of 68 lesions were detected, of which 53 were identified by both MCE and EGD. EGD could more accurately identify ulcers and erosions, whereas MCE showed greater accuracy for identification of erosions and polyps. Only two participants developed transient abdominal pain that spontaneously resolved. The results of this study indicated the safety and effectiveness of this system.

Previous research on the use of NaviCam introduced by Ankon Technologies was reported only by pilot studies and included a limited number of participants; therefore, Liao et al. [19] investigated 350 patients across seven centers to compare the accuracy of MCE with conventional EGD. Using the same procedure that was described by previous studies, this study analyzed the sensitivity and specificity of MCE for detection of lesions across different locations in the stomach. The diagnostic accuracy of MCE was 93.4% (95% CI 90.83–96.02), sensitivity was 90.4% (95% CI 84.7–96.1), and specificity was 94.7% (95% CI 91.0–97.5). Among 104 histopathological lesions detected by conventional EGD, MCE accurately detected early gastric cancer in one patient but failed to detect submucosal tumors (SMT), ulcers, and polyps in two, three, and two patients, respectively. Fundal lesions were more likely to be missed. Among 350 patients, only five (1.4%) reported adverse events, including nausea, abdominal distension, headache, vomiting, and foreign body sensation, all of which were primarily associated with inadequate gastric preparation. Based on this large-scale study using MCE, the authors suggested the utility of this approach for cancer screening, because patients with negative findings on MCE would not need to undergo conventional EGD for further evaluation. However, patients with suspected neoplastic lesions need to undergo conventional EGD as biopsy can only be done with EGD.

Qian et al. [33] performed a similar study in 10 patients who underwent endoscopic submucosal dissection for gastric neoplasia and analyzed the diagnostic accuracy of MCE for the detection of neoplasms. Patients ingested 400 mg of simethicone and approximately 1 L of water before the MCE. Appearance, size, location, and histopathological features of lesions were compared between MCE and conventional EGD. MCE successfully identified all 10 suspected neoplastic lesions as abnormal findings that required further histopathological confirmation. EGD detected 12 focal lesions classified as superficial neoplasms, and MCE missed 1 and detected 11 lesions. The missed lesion was subsequently identified as a tubular adenoma with high-grade dysplasia located at the cardia. All lesions detected by the MCE were located at angle or antrum, the distal parts of the stomach, thus accuracy of detection could be different if more lesions in the proximal stomach were included. Although EGD is the gold standard for diagnosis of gastric neoplasms, lesions can be missed due to incomplete mucosal observation, inadequate gastric preparation, and procedures performed by inexperienced endoscopists [47,48,49,50]. The authors concluded that methods to maximize control and optimal gastric preparation are essential to improve the accuracy of MCE.

Zhao et al. [34] investigated the role of MCE as a potential screening tool for gastric cancer in 3182 asymptomatic individuals who underwent MCE for screening of gastric cancer. Endoscopic biopsy was performed to confirm the lesion in cases of suspected malignancy. Endoscopic treatment was performed in participants with polyps or SMT, and patients with benign ulcers were treated with proton pump inhibitors and underwent 2-month follow-up MCE. In this study, seven patients were diagnosed with advanced gastric cancer, 331 (10.4%) had polyps, 156 (4.9%) had ulcers, and 115 (3.6%) had SMT. All patients swallowed the capsule without difficulty and the mean duration of the examination was 21 min. In this study, the authors used ESNavi, a cloud-based remote reading system developed by Ankon Technologies, which facilitates reading, data storage, sharing, and remote MCE. Experienced endoscopists evaluated the images uploaded on this system from all participating centers. Complete examination was possible using the standard protocol, which suggests the potential role of MCE in remote areas in which patients do not have access to the services of experienced endoscopists. However, EGD was only performed for patients with suspected malignancies and ulcers detected from WCE, thus confirmation of all findings by WCE or possibly missed lesions could not be confirmed by EGD. In addition, patients with suspected neoplastic lesions need to undergo conventional EGD as biopsy can only be done with EGD.

Previous studies have investigated the effectiveness of MCE as a screening and diagnostic tool; Hu et al. [35] investigated the feasibility of the same system in patients at a high risk of EGD-induced bleeding, perforations, and myocardial infarction. The study included 42 patients with a history of temporomandibular joint dislocation during previous EGD, patients with uncontrolled hypertension (blood pressure > 200 mmHg), angina, cerebral infarction, respiratory failure, abdominal pain, and abdominal distension. The results showed that MCE was successfully completed in all patients without any adverse events, the mean time for visualization of the stomach was approximately 28.5 min, and gastric lesions could be observed for a longer time without discomfort to patients. This study showed that MCE was safe and effective in patients in whom conventional EGD was contraindicated owing to the risk of EGD-induced complications.

Similarly to a wire-controlled device in a hand-held magnet [27], Chen et al. [36] developed an additional detachable string attached to one end of the capsule and tested its function in 25 healthy participants who underwent MCE. The string facilitated better controlled movement of the capsule, with minimal-to-no discomfort because the string could be detached at the end of the examination and capsule is moved passively through the GI tract.

Although previous studies have shown that MCE is a useful tool for visualization and identification of gastric lesions, the observation time was longer than that required for conventional EGD, and frame rates, image resolution, and the battery life required improvement. Therefore, in 2019, Ankon Technologies developed a second-generation capsule with an improved frame rate (8 frames/s), a wider view over 150°, improved resolution of 720 × 720 pixels, and an extended battery life. Jiang et al. [37] compared the performance of this novel device with that of a first-generation capsule in a study of 80 patients with GI symptoms; patients were randomly assigned to undergo first- or second-generation capsule evaluation. Maneuverability, image quality, stability, tolerability, and procedural time were compared. The second-generation capsule captured a greater number of frames in the esophagus (171 vs. 2) and showed better image quality (8.63 vs. 7.9) and better maneuverability than the first- generation MCCG (*p* < 0.1). The mean gastric examination time was lower (5.27 min vs. 7.78 min, *p* < 0.001). The second-generation capsule appears to be a promising innovation over previously described devices, which were associated with a longer examination time. Studies on robotic arms have been summarized on Table 4.

## 6. Protocols of Magnetically Controlled Capsule Endoscopy

Increasing experience with these devices has improved image quality, maneuverability, control, and image visualization; however, it was important to address the concerns associated with evaluation protocols, such as methods of gastric preparation and position changes. Some studies administered > 1 L of water prior to the procedure and others used simethicone (defoaming substance) or gas-forming medication to achieve gastric distension. An appropriate preparation protocol is important for successful MCE because persistent bubbles and gastric mucus and insufficient stomach distension can interfere with complete and accurate examination. Zhu et al. [38] performed a randomized, physician-blinded controlled trial to determine the optimal standardized gastric preparation method in 120 patients who randomly received (a) 1 L of water, (b) 950 mL of water and 400 mg of simethicone, or (c) 900 mL of water, 400 mg of simethicone, and 20,000 IU Pronase granules combined with 1 g of sodium-bicarbonate. The image quality was compared based on evaluation of gastric mucosal cleanliness and visualization across different areas of the stomach. Cleanliness of the distal stomach (angulus, antrum, and pylorus) was excellent in all three groups; however, cleanliness of the fundus was good only in the groups that received simethicone and water. Administration of simethicone significantly improved gastric cleanliness (*p* < 0.0001) and Pronase failed to improve the image quality (*p* = 1.00). The results were similar for visualization; simethicone administration significantly improved overall visualization (*p* < 0.0001). Based on these findings, the authors recommended the administration of simethicone with water prior to MCE as the optimal method for gastric preparation, and several subsequent studies have adopted this preparation method.

In a single-blind study, Wang et al. [39] investigated the effects of pre-procedural position changes to improve gastric cleanliness prior to MCE; 83 patients were randomly categorized into study and control groups. Patients in the control group ingested 5 g of dimethicone (antifoaming agent) and walked freely for 15 min before MCE, whereas those in the study group ingested the same amount of dimethicone but changed positions from supine to left lateral, prone, and right lateral positions. The gastric cleanliness score (24 = perfect cleanliness and 6 = poor cleanliness) was graded for each group. The authors observed that patients who changed positions prior to MCE showed significantly greater cleanliness (21.2 vs. 18.6, *p* < 0.001) and concluded that frequent position changes after dimethicone ingestion improved visualization by increasing mucosal contact and prolonging exposure to the antifoaming agent.

Patients’ position during MCE also required attention. Certain patient positions facilitated magnet-controlled capsule movement. Qian et al. [40] investigated the optimal patient position for MCE using the NaviCam system (Ankon Technologies, Co. Ltd., Wuhan, China). Based on a study that included 60 patients, the authors reported that the supine position facilitated the best visualization of the cardia (91.7%, 95% CI 84.6–98.7%, *p* < 0.001) and body (86.7%, 95% CI 78.1–95.3%, *p* < 0.001). The left lateral position was associated with best visualization of the fundus (91.7%, 95% CI 84.7–98.7%, *p* < 0.001), and the knee–chest position was the best for visualization of the angulus (80.0%, 95% CI 69.1–90.1%, *p* < 0.001). The right lateral and sitting positions provided best visualization of the antrum (88.3%, 95% CI 80.2–96.5% and 90.0%, 95% CI 82.4–97.6%, respectively, *p* < 0.001). The right lateral position was also recommended for observation of the pylorus (81.7%, 95% CI 71.5–91.5%, *p* < 0.001). Sun et al. [41] performed a double-contrast barium study of the stomach and spiral CT for virtual anatomical modeling of the stomach prior to MCE. Based on the images obtained, they determined patient positions that enabled optimal visualization of different areas of the stomach and the distance to the ventral surface for optimal placement of the external magnet. The authors observed that stomach visualization using MCE was best achieved in the standing position, followed by the right lateral position, because these positions provided less resistance to capsule movement. Additionally, they observed that the distance between the ventral surface was the shortest for the stomach body and antrum and greatest for the fundus and, therefore, proposed that the anterior abdominal wall is the optimal site for placement of the external magnet to observe the gastric body and antrum, and the left lateral lower chest was the optimal site for fundal visualization.

Another study performed by Cheng et al. [42] investigated the role of a different type of robotic system (Figure 1d). This device also uses a guidance magnet robot similar to NaviCam controller; however, in contrast to previously described systems, it is positioned vertically, similar to conventional chest radiography machines. The external guidance magnet can move horizontally, vertically, or it can rotate, and the patient stands in an upright position while the magnet moves to steer the capsule. No patient reported any discomfort and all 31 patients had good gastric distension. Contrary to difficulty with capsule manipulation at the fundus and cardia associated with previous models, this study reported that the ease of capsule manipulation at the fundus and cardia was similar to that of manipulation at the antrum and body. Patients ingested a large volume of water prior to MCE. Remaining in the supine position after ingestion of a high quantity of water may cause discomfort, which can be avoided by performing the examination in the standing position. This novel method is more comfortable for visualization of the stomach using magnetic steering. Another study by Lai et al. [43] used a standing-type system manufactured by JIFU Medical Technologies (Shenzhen, China) in 161 patients across three centers, who underwent MCE for upper abdominal complaints and compared their findings with those of standard EGD performed 4 h after MCE. No significant difference was observed between two modalities for identification of gastric lesions (*p* = 0.74); three (1.8%) patients reported discomfort, including nausea, oral pain, and dizziness, all of which were associated with gastric preparation procedures. Notably, 60% of the patients indicated that they preferred an MCE over EGD for subsequent evaluation. Studies related to protocols have been summarized on Table 5.

## 7. Future Perspectives of Capsule Endoscopy

Technological advances and evidence-based research over the last decade have contributed to improved maneuverability during MCE performed for visualization of the stomach. Among different types of available magnetic control systems, hand-held devices offer the advantage of portability and the robotic arm system offers a simple and easy method of examination, and it is the most widely available and studied system for MCE of the stomach. Many studies have compared diagnostic accuracy of MCE with EGD and the results are very promising. MCE is better tolerated, and portable devices facilitate evaluation even in remote areas with limited access to specialized hospital services; these features serve as advantages of MCE over conventional EGD. Studies have shown that it is a safe and effective modality to visualize the stomach, and its performance is comparable with that of conventional EGD. Some MCE systems are currently used in clinical practice; however, a few concerns require attention for more widespread acceptability. Currently, no standardized protocols are available for different types of MCE, and the methods of gastric preparation, various pre-CE evaluation protocols, patient positions, and performance assessment methods differ across the various studies that have discussed this subject in the literature. Further large-scale studies are warranted to determine the optimal preparation, procedure protocols, and performance to establish standard guidelines for MCE to promote wider acceptability of MCE. The current system is unable to provide tissue confirmation and histopathologic confirmation is the vital step in diagnosing neoplastic lesions. Further EGD is required for obtaining tissue samples, which limits the use of MCE to simple observation of stomach. Additional procedures, such as polypectomy, endoscopic submucosal dissection, and bleeding control, can only be done through conventional endoscopes. Research should focus on improved methods to obtain tissue samples to encourage widespread clinical application of MCE. Examination time is another problem that needs to be improved. Average time taken for MCE is about 20 to 30 min, which is longer than average of 6 min spent on EGD. Better stomach expansion devices may help reduce the examination time. Notably, MCE is relatively more expensive than conventional EGD ($581.51 [MCE] vs. $145.38 [EGD under sedation]) in China [43]. Further research should focus on cost-benefit analysis and the development of cost-effective measures for this purpose.

## Figures and Tables

**Figure 1 diagnostics-11-01792-f001:**
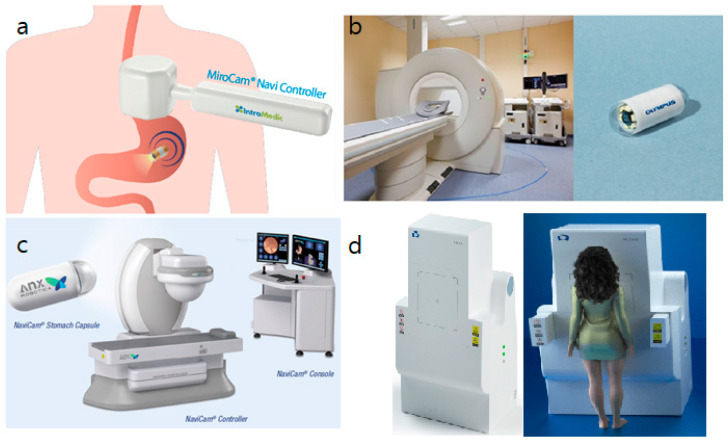
Different magnetically controlled capsule endoscopy systems used in clinical studies: (**a**) hand-held MiroCam-Navi system by Intromedic Ltd. (Seoul, Korea) http://www.intromedic.com, 25 September 2021, (**b**) MCE system developed by Olympus Corp. (Tokyo, Japan) and Siemens Healthcare (Erlangen, Germany) https://www.olympus-global.com, 25 September 2021, (**c**) NaviCam system by AnX Robotica (previously Ankon Technologies, Wuhan, China) https://www.anxrobotics.com, 25 September 2021, (**d**) standing-type MCE by JIFU Medical Technologies (Shenzen, China) http://www.jifu-tech.com, 25 September 2021.

**Table 1 diagnostics-11-01792-t001:** Systems of MCE used in clinical studies.

Hand-Held System	MRI System	Robotic Arm System
Mirocam-Navi (Intromedic Ltd.)	Olympus Corp and Siemens Healthcare	NaviCam (AnX Robotica, previously Ankon Technologies)
InsightEyes EGD system (Insight Medical Solutions, Taiwan)		OMOM robotic capsule endoscopy (Jinshan Science & Technology)

**Table 2 diagnostics-11-01792-t002:** Clinical studies using magnetic guided capsule gastroscopy controlled by hand-held magnet.

Author	Participants	Capsule	Observation Time	Results
Swain et al. [22]	1 healthy volunteer	Colon capsule (Given Imaging Ltd.)	15 min	Successful manipulation of MCE, no adverse events.
Keller et al. [23]	10 healthy volunteers	Colon capsule (Given Imaging Ltd.)	39 ± 24 min	75% to 90% visualization in 7 patients, no adverse events.
Rahman et al. [24]	26 healthy volunteers	MiroCamNavi (Intromedic Ltd.)	24 min (12–39 min)	100% visualization in body, 88% in cardia, no adverse events.
Ching et al. [25]	49 patients with IDA	MiroCamNavi (Intromedic Ltd.)	23 min	More lesions detected by MCE than EGD (36 vs. 5, *p* < 0.001).
Ching et al. [26]	33 patients with suspectec UGI bleeding	MiroCamNavi (Intromedic Ltd.)	20 min	No difference in significant lesions detected by MCE than EGD (14 vs. 13, *p* = 1).
Lien et al. [27]	9 healthy volunteers	Cable capsule developed by authors	27.1 min	85.2% of stomach examined clearly, no adverse events.
Lin et al. [28]	15 home care patients	InsightEyes^®^ EGD (Insight Medical Solutions Corp)	23.7 ± 10.0 min	81.25% of stomach landmarks observed with detail, no adverse events.

IDA, iron deficiency anemia; MCE, magnetically controlled capsule endoscopy; EGD, esophagogastroduodenoscopy.

**Table 3 diagnostics-11-01792-t003:** Clinical studies using magnetic guided capsule gastroscopy controlled by an electromagnetic coil system.

Author	Participants	Capsule	Observation Time	Results
Rey et al. [29]	29 healthy volunteers, 24 patients with UGI symptoms	Capsule endoscope (Olympus)	30 min (8–50 min)	Visualization up to 98% in antrum and 73% in fundus; 14 lesions detected by capsule and EGD (10 vs. 6). No adverse events.
Rey et al. [30]	61 with UGI symptoms	Capsule endoscope (Olympus)	17.4 min (9.9–26.4 min) for MCE, 5.3 min (4.4–6.3 min) for EGD	Visualization up to 93.4% in body and 85.2% in fundus. 108 lesions detected by capsule and EGD (94 vs. 77). No adverse events.
Denzer et al. [31]	189 patients with UGI symptoms	The PillCam ESO2 (Given imaging)	10.6 min (10.1–11.1) for CE, 4.0 min (3.7–4.2) for EGD	23 major lesions. Accuracy 90.5%, specificity 94.1%, sensitivity 61.9%.

UGI, upper gastrointestinal; EGD, esophagogastroduodenoscopy; MCE, magnetically controlled capsule endoscopy; CE, capsule endoscopy.

**Table 4 diagnostics-11-01792-t004:** Clinical studies using magnetic guided capsule gastroscopy controlled by robotic arm.

Author	Participants	Capsule	Observation Time	Results
Liao et al. [32]	34 healthy volunteers	AKE-1 (Ankon Technologies)	43.8 ± 10.0 min	Up to 95% gastric mucosa observed clearly, accurate movement in 85.3%. No adverse events.
Zou et al. [20]	68 patients with UGI symptoms	NaviCam (Ankon Technologies)	29 min	68 lesions detected (52 by MCE and 50 by EGD). Overall agreement of 91.2%.
Liao et al. [19]	350 patients with UGI symptoms	NaviCam (Ankon Technologies)	26.4 ± 5.1 min	For detection of gastric lesions, sensitivity 90.4%, specificity 94.7%.
Qian et al. [33]	10 patients with known superficial neoplasia	NaviCam (Ankon Technologies)	N/A	Per-patient sensitivity 100%, per-lesion sensitivity 91.7%.
Zhao et al. [34]	3182 asymptomatic individuals	NaviCam (Ankon Technologies)	21 min	Detected 7 AGC, 145 ulcers, 319 polyps, and 114 SMTs.
Hu et al. [35]	42 high risk patients	NaviCam (AnX Robotica, previously Ankon Technologies)	28.5 min	No adverse events.
Chen et al. [36]	25 healthy volunteers	Navicam (Ankon Technologies) with detachable string	14.3 min	No adverse events.
Jiang et al. [37]	80 participants	Second generation capsule (AnX Robotica, previously Ankon Technologies)	5.27 ± 0.74 min	Higher image quality (8.63 vs. 7.9), shorter examination time (5.27 min vs. 7.78 min, *p* < 0.001).

UGI, upper gastrointestinal; N/A, not applicable; MCE, magnetically controlled capsule endoscopy; EGD, esophagogastroduodenoscopy; AGC, advanced gastric cancer; SMT, submucosal tumor.

**Table 5 diagnostics-11-01792-t005:** Clinical studies on protocols of magnetically controlled capsule endoscopy.

Author	Participants	Control Device	Compared Procedure	Results
Zhu et al. [38]	120 patients with UGI symptoms	Robotic arm	Comparison of gastric preparation method.	Visualization is improved with use of simethicone (*p* < 0.0001).
Wang et al. [39]	83 patients with UGI symptoms	Robotic arm	Benefit of repetitive position change prior to MCE.	Visualization is improved with use of dimethicone and repetitive position change (*p* < 0.001).
Qian et al. [40]	60 patients with UGI symptoms	Robotic arm	Optimal position for different parts of stomach.	Left lateral for fundus, knee-chest position for angulus, right lateral and sitting position for antrum, and right lateral position for pylorus.
Sun et al. [41]	10 healthy volunteers	N/A	Optimal patient position and magnet location.	Standing position provides less resistance to capsule movement. Place magnet at anterior abdominal for observing body and antrum, left lateral lower chest for fundus.
Cheng et al. [42]	31 healthy volunteers	Robotic arm	Feasibility of standing position.	Standing position was well tolerated with good visualization.
Lai et al. [43]	161 patients with UGI symptoms	Robotic arm	Feasibility of standing position.	Similar diagnostic accuracy to EGD, well tolerated.

UGI, upper gastrointestinal; N/A, not applicable; EGD, esophagogastroduodenoscopy.

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
