# Peer review of "Capsule Endoscopy for Gastric Evaluation"

_diagnostics, 2021, doi:10.3390/diagnostics11101792_

Round 1

Reviewer 1 Report

This Review describes the entire field of magnetic controlled gastric capsule endoscopy by comprehensively discussing all types of System, principles of performance and outcomes. The Contents aids significantly to this Special issue on development of capsule endoscopy and to scientific interst in General.

Some points are suggested:

It should be clearly defined and discussed what the Terms: wireless capsule endoscopy, capsule endoscopy, magnetic controlled CE, Magnetically assisted CE, Magnetically  guided CE stand for. Is this terminology coming from different Producers (MiroCam vs. NaviCam) or is there a real medical or technical reason?  A table might be helpful, including subtypes of hand-held magnet, robotic arm, string. 

Are there 3 types / generations  of Ankon capsules?

Ankon  is now acting as AnX robotics

In the text a FUZI and a JIFU capsule are described. The reader might think of two different types. However, when looking into the 2 cited papers, the images of the System look identical.

It should be mentioned which Systems are available presently

The Impact of this review would be significantly increased when including images of the different Systems.

Author Response

Comment 1) It should be clearly defined and discussed what the Terms: wireless capsule endoscopy, capsule endoscopy, magnetic controlled CE, Magnetically assisted CE, Magnetically guided CE stand for. Is this terminology coming from different Producers (MiroCam vs. NaviCam) or is there a real medical or technical reason? A table might be helpful, including subtypes of hand-held magnet, robotic arm, string.

Answer: Thank you for your comment. As you have pointed out, terminology can be confusing to readers who are not familiar with the system. Currently, there is no specific terminology for different magnetic systems. The AnX Robotica Corporation (previously ANKON medical technologies) adapts term “magnetically controlled capsule endoscopy system”. Studies have also used terms such as magnetically controlled capsule endoscopy (MCE) and magnetically controlled capsule gastroscopy (MCCG). Olympus Medical Systems Corporation and Siemens Healthcare used the term “magnetically guided capsule endoscopy system”. Meanwhile IntroMedic describes their device as “capsule endoscope that can control the movement of capsule using Magnetic Controller”, and a study by Rahman et al used the term “magnetic assisted capsule endoscopy (MACE)”. So some terms have been used by the manufacturers, and some terms have been used by authors. In order to unify terminology, we have used the term magnetically controlled capsule endoscopy (MCE) as general term to describe all systems using magnets for external actuation.

We have added a figure with representative types of MCE.

Are there 3 types / generations of Ankon capsules?

The most recent capsule by AnX robotica (previously ANKON medical technologies) uses NaviCam® capsule with resolution at 640 x 480 pixels and frame rate up to 12 fps. First generation provided image resolutions of 480 x 480 pixels with frame rate at 2 fps, and the second generation provided image resolutions of 720 x 720 pixels with frame rate at 8 fps.

Ankon is now acting as AnX robotics

Thank you for your comment. China based ANKON Medical Technologies was established in 2009, and commercial agreement was established in 2019 with U.S. based AnX Robotica to share products and technologies with ANKON. We have changed the name as AnX Robotica (previously ANKON technologies) for studies reported after 2019.

In the text a FUZI and a JIFU capsule are described. The reader might think of two different types. However, when looking into the 2 cited papers, the images of the System look identical.

Thank you for your comment. We were aware of this point, as reference 45 studied performance of standing type magnetically guided capsule endoscopy developed by JIFU Medical Technologies Co., Ltd., but reference 44 discusses same standing type with same illustration but different company FUZI Technologies Co. LTD (Shenzhen, China). There could a typing error from the authors in reference 44, and there is actual evidence that JIFU medical technologies have developed and marketed standing type magnetically guided capsule endoscopy, thus the name from reference 44 was omitted.

It should be mentioned which Systems are available presently

Currently available systems are magnetically controlled capsule endoscopy system by AnX Robotica, OMOM robotic capsule endoscopy system by Jinshan Science & Technology (Group) Co., Ltd., standing type magnetically controlled WCE developed by JIFU Medical Technologies Co., Ltd. (Shenzhen, China), MiroCan Navi (Intromedic Ltd, Seoul, South Korea), InsightEyes EGD system (Insight Medical Solutions, Taiwan) and Olympus Medical Systems Corporation and Siemens Healthcare. We have added a table and images of different devices. We added table 1 to summarize current systems.

The Impact of this review would be significantly increased when including images of the different Systems.

Thank you for your comment. We have added figure 1 to help readers understand different systems better.

Reviewer 2 Report

Dear colleagues,

regarding your paper "Capsule Endoscopy for gastric evaluation":

This is a well structured article regarding an increasingly important subject, as modern medicine heads to a minimally invasive approach in different fields.

An extensive review was performed and adequately compiled, and is useful for both CE experts and other endoscopists/gastroenterologists.

However, several oversights and incorrections in the paper were found:

1 - English needs a significant overhaul, in particular the Introduction section

2 - Inconsistencies were detected between data presented in text and in tables (e.g. references 21 and 22)

3 - Some data should be analyzed critically, especially regarding major differences between CE and EGD diagnostic yield across different authors using similar devices - a closer look at local expertise and definition of lesions could explain these differences

4- In the very important study by Denzer et al, [28], it would be important to note that high accuracy does not make up for a low sensitivity for major lesions

5 - A conclusion on page 8 regarding utility of MCE for gastric cancer screening should take into account that the majority of guidelines require histopathologic staging for the definition of gastric cancer screening, impossible to obtain with MCE (at the moment).

6 - A diagram regarding the different magnetic systems would be interesting to present, to better understand how the mechanism works

7 - In my opinion, the major shortcoming of this article is the (unconscious?) cherry-picking regarding MCE. It is a wondrous development, with exciting prospects, but a critical view should be employed at all times. Comparing tolerability and diagnostic yield of MCE versus unsedated EGD, and further on, costs versus EGD with sedation is a bias that should be avoided or at least emphasized. Moreover, a section regarding MCE limitations is obligatory, including procedure duration  (significantly increased versus EGD), visibility issues and lack  of tissue sampling. Finally, considering that off site reading could benefit remote locations is hard to accept since an expert in MCE is still needed on site (this is in sharp contrast to SBCE and CCE that don't require experts on site in the vast majority of the circunstances).

In my opinion, the manuscript should be revised in order to better reflect the subject at hand.

Author Response

1 - English needs a significant overhaul, in particular the Introduction section

Thank you for your comment. The english editing was done by Editage, and we have attached the certificate of english editing. We have looked over the entire manuscript meticulously and made adjustments to introduction as well as the entire manuscript.

2 - Inconsistencies were detected between data presented in text and in tables (e.g. references 21 and 22)

Thank you for your comment. Observation time for reference 21 was revised, and values were revised to include gastroscopy findings only. Also values for reference 22 was revised to include all significant lesions.

3 - Some data should be analyzed critically, especially regarding major differences between CE and EGD diagnostic yield across different authors using similar devices - a closer look at local expertise and definition of lesions could explain these differences

Thank you for your comment. Some parts were revised, and the corrected parts are highlighted under “track changes option”.

4- In the very important study by Denzer et al, [28], it would be important to note that high accuracy does not make up for a low sensitivity for major lesions

Thank you for your comment. We have added the following comments.

Although WCE in detecting major lesions had high specificity of 94.1%, the sensitivity of WCE was only 61.9% which suggests major improvement is needed before using it as a screening test to identify patients requiring further EGD. One of the major limitations of this device was limited expansion of stomach. Unlike conventional EGD that uses air for stomach expansion maintenance of expanded state, the water ingested during WCE left stomach too quickly hindering accurate examination of single focal lesions.

5 - A conclusion on page 8 regarding utility of MCE for gastric cancer screening should take into account that the majority of guidelines require histopathologic staging for the definition of gastric cancer screening, impossible to obtain with MCE (at the moment).

Thank you for your comment. This is an important factor that needs to be emphasized, thus we have added the comment “However, patients with suspected neoplastic lesions need to undergo conventional EGD as biopsy can only be done with EGD.” Also, we have made adjustments to the “future perspectives of capsule endoscopy” section.

6 - A diagram regarding the different magnetic systems would be interesting to present, to better understand how the mechanism works

Thank you for your comment. We have received similar comment from reviewer 1, thus we have added Figure 1 showing examples of some systems available, and Table 1 showing commercially available systems.

7 - In my opinion, the major shortcoming of this article is the (unconscious?) cherry-picking regarding MCE. It is a wondrous development, with exciting prospects, but a critical view should be employed at all times. Comparing tolerability and diagnostic yield of MCE versus unsedated EGD, and further on, costs versus EGD with sedation is a bias that should be avoided or at least emphasized. Moreover, a section regarding MCE limitations is obligatory, including procedure duration (significantly increased versus EGD), visibility issues and lack of tissue sampling. Finally, considering that off site reading could benefit remote locations is hard to accept since an expert in MCE is still needed on site (this is in sharp contrast to SBCE and CCE that don't require experts on site in the vast majority of the circunstances).

Thank you for your valuable comment. Although commercially available, the MCE still needs a lot of issues and technical problems that have to be dealt with. We have added such issues in “future perspectives of capsule endoscopy”.

Round 2

Reviewer 2 Report

The changes performed to the paper, both suggested by me and others you have performed on your own, have greatly improved the article's quality and objectivity.

Congratulations on your relevant work in this field!